# Pyrolysis and Torrefaction—Thermal Treatment of Creosote-Impregnated Railroad Ties as a Method of Utilization

**DOI:** 10.3390/ma16072704

**Published:** 2023-03-28

**Authors:** Paweł Kazimierski, Paulina Kosmela, Piotr Piersa, Szymon Szufa

**Affiliations:** 1The Szewalski Institute of Fluid-Flow Machinery, Polish Academy of Sciences, Fiszera 14, 80-231 Gdansk, Poland; 2Department of Polymer Technology, Faculty of Chemistry, Gdansk University of Technology, G. Narutowicza 11/12, 80-233 Gdansk, Poland; 3Faculty of Process and Environmental Engineering, Lodz University of Technology, Wolczanska 213, 90-924 Lodz, Poland

**Keywords:** pyrolysis, railroad ties, waste, creosote, Soxhlet extraction

## Abstract

A fundamental issue of waste management and the rail transport industry is the problem of utilizing used railroad ties. Wooden railroad ties are treated with a preservative, usually creosote. Due to their high toxicity, railroad ties are considered hazardous waste and must be utilized under various directives. It is proposed to utilize the troublesome waste by using the pyrolysis and torrefaction process. The research proves that the thermal method is effective for disposing of this type of waste. Torrefaction up to 250 °C gives high efficiency of impregnation removal, while pyrolysis up to 400 °C completely neutralizes waste. A series of experiments were conducted for various final pyrolysis temperatures to determine a minimum temperature for which the obtained solid products are free from creosote. Extraction with the use of the Soxhlet technique was performed for the raw materials and the obtained solid products—chars. The oil content for liquid fraction was also examined for each sample. As a result of the thermal treatment of the waste, fuel with combustion parameters better than wood was obtained. For a high final temperature of the process, the calorific value of char is close to that of hard coal.

## 1. Introduction

Impregnated wooden railway tie waste constitutes a significant stream of hazardous waste amounting to 120 k tons per year in Poland and a few times more (0.6–1.5 mln tons) in Europe. Additionally, in connection with planned renovations of the railways in Poland, these amounts may increase significantly [1]. Despite the tendency to replace wooden ties with ties made of concrete and the numerous research projects focused on finding other composite materials for tie production [2,3,4], the problem of disposal of old ties installed a few decades ago and dismantled nowadays is recent. This will probably be problematic for the next several dozens of years since a small portion of railroads is still built with wooden ties impregnated with creosote. Elements made of wood are used for both technical and economic reasons. Wooden railway ties are much more resistant to temperature fluctuations and are better vibration absorbers. They are also cheaper than concrete ties, an economical issue that is usually an essential aspect when deciding on which material will be used for construction. Experimental studies [5] performed on a materials testing machine showed that the compressive strength of wooden railroad ties in the middle part of the tie is higher than for the prestressed concrete ties.

Wooden ties can be impregnated using surface methods or, much more often, soaking methods [6]. The second method (using high pressure) has been widely used up to now, and this method was developed in 1902 by Rumping and Wasserman [7]. This method consists of placing the wooden tie inside a cylindrical container, inside which the pressure is raised up to 3–5 atm depending on the wood species. The temperature of soaking is 90 °C. Then, the pressure is increased to 6–8 atm to soak in the total volume of the tie. When the wood structure is saturated with the impregnate, the pressure is dropped back to the atmospheric pressure. The pressure difference results in the removal of a portion of the preserving oil from the whole volume of the tie. This method provides a good saturation of the whole volume of the wood with low impregnating oil consumption [8]. The impregnation of a wooden railroad tie is essential in its production. A non-preserved tie (e.g., made from beech wood) has a durability of approximately 3 years, while an impregnated one has30 years. Besides this obvious benefit of wood impregnation, the other aspect is using creosote oil, which is a toxic waste product [8]. However, during the time of exploitation, it is approximated that approximately 5 kg of creosote is emitted, which gives the emission factor of 208 mg/m^2^ per day [9]. This example assumes that one railroad tie with approximately 15 kg of creosote oil, and during a service life of 30 years, will have approximately 5 g of oil washed or evaporated from it.

Creosote oil (an impregnation) is a mixture of coal or wood tar; it is a product of a distillation process in the range of 200–360 °C. It consists mainly of PAHs (polycyclic aromatic hydrocarbons), such as naphthalene, anthracene, phenanthrene, pyrene, and chrysene. Acid components constitute 4 to 16% (mainly phenols) and alkaline components 3.5–4.5% (pyridine, quinolone) of the creosote oil. The difficulty of estimating the quantity of the creosol oil in railway tie waste is a lack of a precise standard describing the quantity of oil absorbed by the wood during the impregnation process (e.g., PN-EN 13145+A1:2011 does not describe the quantity of oil in the ties) [10]. The amount of absorbed oil varies regarding the wood species and specific application of the ties, and it was described in a withdrawn standard (PN-D-95014) [11], and it ranged between 45 kg per m^3^ of wood up to 150 kg/m^3^. The number of installed railway ties in Poland is gathered in Table 1. This paper proposes the method of torrefaction and pyrolysis as a method of waste disposal. Physical properties, specifically the distillation temperature, indicate the possibility of the physical removal of impurities by evaporation. However, the authors suggest a higher process temperature to remove the impurities remaining in the wood through their decomposition.

Lin et al. [13] analyzed the feasibility of using used wooden railroad ties impregnated with creosote for use in exterior structural applications regarding its mechanical properties. However, this solution, although widely used, is not suitable. It disperses waste instead of disposing of it.

Gonzalez et al. [14] performed an experimental investigation on the possibilities of using biochar from creosote-impregnated wooden ties for soil amendment. They conducted pyrolysis at 667 and 700 °C, and the char produced with higher temperatures had lower PAH content. The proposed utilization method uses pyrolysis with the simultaneous production of material useful for many different industrial sectors, e.g., a smokeless solid fuel, just like other types of waste biomass [15,16,17,18]. There are also other types of railroad valorization methods, one of which is called torrefaction, which is thermochemical conversion in the absence of oxygen. One of the atmospheres in which railroad ties can be carbonized is superheated steam production technology with simultaneous pressure and temperature control of SHS technique, which is cheaper in industrial applications than other types of torrefaction processes that are using nitrogen or argon. The solution is effective; however, the proposed heat treatment temperature is troublesome. Pyrolysis reactors operating at such temperatures must be made of expensive materials. Due to the cost of materials and lower thermal efficiency of processes carried out at higher temperatures, the authors, apart from pyrolysis, also check torrefaction as a disposal method. Thermal treatment of used creosote-impregnated ties at temperatures between 250 and 350 °C was adopted by Kim et al. [15]. Their analysis showed that the amounts of creosote removed from the samples ranged between 47 to 79% of the amount in the raw material. Similar to the previous research, the lowest PAH content was obtained for the highest process temperature (97% removed). Kim et al. [15] used torrefaction as a method of disposal; however, the paper indicated incomplete neutralization of the waste, which is why the authors decided also to use a higher temperature.

Reactor design of the torrefaction process includes: lack of economically/environmentally efficient technological options for reactor heating and pre-drying of biomass and use of “torgas” technology combined with fossil fuels. Current reactor technologies are inefficient as they require a long biomass residence time in the reactor. Innovative superheated steam (SHS)-powered torrefaction reactor includes: improving process control of temperature, torgas production, and residence time, reducing residence time and emissions, increasing the recovery rate of by-products from condensate, improving thermal efficiency (heat recovery during water re-condensation), and allowing the use of semi-dry biomass residues (moisture content up to 60%) [19,20,21]. By-products from the torrefaction process include: limited knowledge of valorization options, particularly for the wastewater from the condensate, which is rich in volatile organic compounds and valorization of the condensate by-product for biofuel production (biogas, biohydrogen) and recovery of bioproducts [22].

Summarizing the research ideas, the authors use the thermochemical conversion of railroad ties to dispose of them. The experiments used temperatures at which both torrefaction and pyrolysis occur. The use of lower process temperatures has significant advantages, such as a larger amount of char that can be used for further energy purposes. In the case of a process carried out at a higher temperature, i.e., in pyrolysis and not torrefaction, the amount of volatile matter decreases with temperature. Reducing the amount of volatiles results in a carbon concentrate free of volatiles [23,24]. The work describes the effectiveness of creosote oil removal from the waste, as well as the fuel properties of the solid product obtained as a result of the thermal process.

## 2. Materials and Methods

### 2.1. Raw Materials and Preparation

The used railroad tie from a rail freight line in Gdańsk, Poland, was utilized for the present study. The experiments were performed on sawdust resulting from drilling through the tie to provide homogeneous samples. The samples were obtained regarding the standard method, PN-EN 351-2:2009 [25]. The drillings were made in a central part of the tie since the edges usually have less oil due to leaching throughout the years of exploitation. Figure 1 presents the cross-sections of the wooden tie through the center of length and 1 cm from the edge. Figure 2 presents the drilling and the obtained raw material sample.

### 2.2. Analysis and Evaluation Methods

The railroad ties were pyrolyzed in a batch reactor. Each process was performed for a 20 g sample in the form of dried sawdust (105 °C). The samples were placed in a reactor and transferred to a heated, high-temperature furnace. The pyrolyzing reactor was provided with two nozzles: one for the inflow of inert gas to purge the sample and the other nozzle for the outflow of pyrolytic gases and liquid fractions, including the creosote oil.

The system of liquid fraction sampling and the impregnate was composed of two gas-washing bottles filled with isopropanol. The inert gas provided to the reactor was nitrogen, and its flow rate was controlled with a peristaltic pump. The flow rate for the conducted experiments was 170 mL/min. Inert gas had no real effect on pyrolysis products, and it served to better remove pyrolysis oil from inside the reactor for better analysis. The experimental investigation incorporated torrefaction and pyrolysis processes of the samples by heating the reactor to 200, 250, 300, 350, and 400 °C and holding these temperatures until the process was over. The obtained char was subjected to the extraction process with the Soxhlet method to evaluate the cleaning process. This is the method used as an extraction technique. The principle of this method is the multiple passing of pure (distilled) solvent through the extracted material. The extract then goes to the distillation flask under the Soxhlet apparatus. The toluene was used as a solvent, evaporated from the flask, and returned to the apparatus for re-extraction. This solution allows for multiple extractions using a small amount of solvent. Then, the obtained extract was analyzed using a chromatograph to evaluate the amount of the impregnate left in the char sample. The char was also subjected to the heat of combustion evaluation to check its fuel usability. Volatile parts and ash were also determined in the raw material and the obtained samples. Experiments were conducted in a setup presented in Figure 3.

In order to analyze the rate of thermal transformations and temperatures in which they occur, thermogravimetric analyses of the waste sample were carried out. This type of analysis makes it possible to determine the optimal conditions process and the dynamics of pyrolysis of various raw materials. The results of the thermal decomposition of waste presented in this work were performed using the SDT Q600 thermogravimeter. The gases generated in the TGA analysis were directed to the spectrometer. A flow-through gas cuvette connected to a thermobalance with a thermostatic transfer line was used in the tests. Gas (nitrogen) flow between TGA and FTIR was 100 mL/min. Interpretation of the obtained spectrum allowed the identification of the gaseous products formed. In the analysis of the spectra, the main emphasis was placed on the intensity of the peaks for wave numbers characteristic of the components of creosote oil (e.g., the occurrence of aromatic compounds).

## 3. Results and Discussion

### 3.1. Thermal Analysis

#### Thermogravimetric Analysis Combined with FTiR

The first stage of the research is the thermogravimetric and qualitative infrared TGA-FTIR analysis. Figure 4 shows the mass loss in temperature function for impregnated wooden tie and the purified (with the Soxhlet method) wooden tie sample. The figure also shows a derivative of mass in the function of temperature, which represents the intensity of the mass loss for the two samples. It can be seen that both samples are characterized by a similar course of the mass loss curve resembling the results for pure wood in the pyrolysis process. After extraction, the sample loses its mass in a less dynamic way, resulting from a lack of a low-boiling component—the impregnate. Since the impregnant is a small waste mass, the graphs follow a similar pattern. However, the charts confirm the content of components leached by toluene, e.g., PAHs.

FT-IR analysis results confirm the presence of the impregnate in the producer gas obtained from heating the samples in an inert gas environment. The result is a base to consider pyrolysis as a proper wooden tie purification method. Figure 5 shows the result of the measurement performed for the purified wooden tie sample. The presence of a gas can be observed from the 17th min of the heating process, corresponding to a temperature of approximately 250 °C. The most intense and the earliest to be observed signals are a sign of the presence of the carbon mono- and dioxide, which is also reflected in the two asymmetrical vibrational bands stretching the bonds of C=O at approximately 2350 cm^−1^ and 2100 cm^−1^. The results are very similar to those obtained for biomass—pure wood. Analysis shows that railroad ties cleaned with creosote oil have the properties of ordinary wood. Dynamic mass loss occurs for temperatures above 270 °C, i.e., as for pure wood.

In the case of an impregnated wooden tie (Figure 6), besides the characteristic bands resulting from the presence of the pyrolytic gas (e.g., 2350 cm^−1^ and 2100 cm^−1^), an intense band indicating the presence of hydrocarbons as the base component of the impregnate can also be observed. This is represented by asymmetrical bands of vibrational stretching and symmetrical vibrational bands for bands C-H of methyl groups at 2960 and 2818 cm^–1^. Another band is observed at 1500–1600 cm^−1^, representing stretching vibrations of aromatic bonds C=C in the evaporating PAH. 

This is evidenced by the presence of intense bands of asymmetric stretching vibrations and symmetrical stretching vibrations of C–H bonds of methyl groups at 2960 and 2818 cm^−1^. There is also a band in the range 1500–1610 cm^−1^, caused by stretching vibrations of aromatic bonds C=C, coming from evaporating from the sample PAHs). Strong signals from aromatic bonds appear already at low temperatures. These signals confirm that creosote oil is released as a result of heat treatment. Studies clearly show that the removal of pollutants occurs both during torrefaction and also at higher temperatures. This indicates that pyrolysis is necessary to purify this waste from PAHs completely.

The extract samples obtained from the chars and the pyrolytic oil were analyzed with the use of a chromatograph. Figure 7 shows pyrolytic oils and extracts from the obtained chars. The figure clearly presents the oil transition from wood to liquid pyrolysis products. The figure shows how the color of the extract from the biomass after thermal treatment changes and how, in inverse proportion, the color of the pyrolytic oil changes. For low final temperatures of pyrolysis, since most of the impregnate was left in the wooden ties, the color of the pyrolytic oil was bright. For higher temperatures, the extract was bright, while the pyrolytic oil was of intense, dark color. The extracts, as well as pyrolytic oils, were subjected to further measurements with the use of gas chromatography (Shimadzu GC-MS).

Figure 8 shows the results of the total PAH content and six main constituents of PAHs in the pyrolytic oils. The results concerning the PAH content in a raw wooden tie sample (extract) are presented. The temperature of 200 °C content of the creosote oil is of trace quantity and amounts to 2.5% of the initial mass of the impregnate. An intense increase in the amount of removed impregnate occurs at 250 °C, where the quantity of the removed impregnates rises to 36%. The value increases with temperature and amounts to 52% for 400 °C. The intense rise of impregnate content between 200 °C and 250 °C results from the boiling point of the impregnate being somewhere in the given range. The temperature of 400 °C (the highest from all the experiments) is much higher than the boiling point of the impregnate. The lack of the entire balance of the creosote oil in the pyrolysis oil and the extract may be due to oil degradation due to high temperature. Long hydrocarbon chains undergo thermal decomposition before they are vaporized from the pyrolysis reactor. This process results in the formation of carbon and secondary pyrolysis gases or secondary pyrolysis liquids with shorter chains. As the process temperature increases, the waste is stripped of more pollutants. After conducting the process at high temperatures, the waste can be considered safe [26,27,28,29].

During the GC-MS analyses, the areas under the surface of all signals obtained at the times for which the PAH reference mixture was obtained were calculated. The amount of chemical compounds was very large; however, comparing them with the six compounds that were the most in the sample, it was found that these six compounds are a representative sample, and the decrease in the concentration of these compounds proved to be a large extent of the decrease in total PAHs. Mikulski et al. describe a standard-based chromatographic analysis method for PAHs [30]. Six main PAHs were analyzed in detail, and they were selected on the basis of a sample of the raw primer extract for which the six main ingredients are: acenaphthylene, dibenzofuran, fluorene, anthracene, pyrene, and fluoranthene. These six compounds were also the predominant components for most of the pyrolysis oil samples (250, 300, and 350 °C). For 200 °C oil, only four components of the oil were determined: acenaphthylene, dibenzofuran, fluorene, and phenanthrene. For the oil obtained at 40 °C, the five main compounds coincided with the main ones from the extraction, and the sixth one is a naphthalene derivative that was found in higher concentrations compared to fluoranthene. The diagram below shows the concentrations of the six main components of the impregnation, and these values were compared with the concentrations obtained from the extraction of the raw substrate. The transition of the impregnation to the pyrolysis oil is advantageous, since in the waste disposal installations, the pyrolysis oil is combusted. As the process temperature increases, the waste is stripped of more pollutants. After conducting the process at high temperatures, the waste can be considered safe. Thermal methods are the most recommended methods of neutralizing this type of waste. Recycling raw materials or materials is impossible due to the oils contained, and energy recycling methods allow for low-cost waste disposal while obtaining energy.

The idea of thermal cleaning of railway sleepers from impregnation (Figure 9) is to prepare them for use for energy purposes. The pyrolysis process is an effective method of their disposal [31,32]. Moreover, the process of low-temperature pyrolysis torrefaction results in the fact that the treated base has better parameters as a fuel compared to the wood from which the base was obtained. This is due to the increased concentration of elemental carbon in the biomass. In order to characterize the waste as a fuel, the calorific value (Figure 10), volatile matter content (Figure 11), and ash content were determined [33,34]. As a result of the loss of volatiles, the amount of ash in the char is higher than in the unprocessed waste. The increase in ash content is proportional to the loss of volatile matter due to pyrolysis.

The raw material and the obtained chars were subjected to LHV analysis. The calorific value of the obtained solid fraction increases with the temperature of the process. It is the result of increasing the concentration of carbon in the product [35,36,37]. In the first phase of the pyrolysis process, taking place at low temperatures of 250–350 °C, carbon monoxide and carbon dioxide are released from the biomass, reducing the oxygen concentration in the biomass, which component in the fuel lowers its calorific value. The raw material was characterized by a heat of combustion of less than 21 MJ/kg; during pyrolysis, this value gradually increased to the level of almost 30 MJ/kg for the char obtained at the temperature of 400 °C. The most significant increase in the heat of combustion is observed in the range of 300–350 °C [38,39].

A reverse trend compared to the increasing heat of combustion was observed in the amount of volatile matter. With temperature, the share of volatile parts decreases, with the most intense decrease observed for the range 300–350 °C. This phenomenon coincides with the thermogravimetric analysis results presented in Figure 5. From the point of view of the nature of the research, the decrease in the amount of volatile matter in the initial phase of sample heating seems to be more interesting [39,40,41,42,43,44]. The difference in the content of volatile parts in the raw material and the torrefaction obtained at 200 °C is 5.4%. Assuming that the raw material used for the tests was dried, this difference results from the beginning of the process of the impregnation vaporization to the temperature of 20 °C. The phenomenon of evaporation of the impregnant already at low temperatures proves the high vapor pressure of the impregnant and is a phenomenon of concern for environmental reasons. Railway sleepers have the potential to emit pollutants (PAHs, the main compounds being pollutants listed in Figure 9) into the air in the form of organic vapors. Dangerous compounds evaporating from the waste are organic compounds that are a component of the impregnant. The natural components of raw wood decomposition do not threaten the environment.

An important parameter of the char is its ash content (Figure 12) from the point of view of using it for energy purposes. The raw railway sleeper had an ash content of about 2.7%, and the pyrolysis char at 40 °C contained more than twice as much ash (6.6%). A series of new types of pyrolysis reactors can be taken under consideration in order to valorize crude railroads [45].

## 4. Conclusions

Pyrolysis and torrefaction are suitable methods of neutralizing railway sleepers. In addition to the neutralization function, this process serves to obtain solid fuel, which, thanks to its thermal treatment, is free from harmful impregnants. The char obtained is a coal concentrate; its calorific value increases with the temperature. Moreover, the process of low-temperature pyrolysis and torrefaction causes the substrate after thermal treatment to have a higher heat of combustion compared to the wood from which the base was obtained. This is due to the increased concentration of elemental carbon in the biomass. Conducting thermal neutralization at a temperature of 400 °C allows for keeping about 75% of the chemical energy of the biomass in the char while at the same time changing the weight of the sleepers by less than 50%. This method allows the production of fuel with a heat of combustion comparable to the heat of combustion of coal, using waste that is difficult to manage. Removal of organic impregnants from railway ties takes place at a temperature of 200 to 400 °C. The most intensive evaporation takes place from 200 to 250 °C, in which most of the impregnation is removed. The torrefaction process is sufficient to remove most impurities from the waste. However, the pyrolysis process is more effective, and the waste subjected to this process guarantees the removal of toxins. An important element of the process of neutralizing sleepers by pyrolysis is the utilization of pyrolysis gases, as they are a carrier of potentially dangerous pollutants.

## Figures and Tables

**Figure 1 materials-16-02704-f001:**
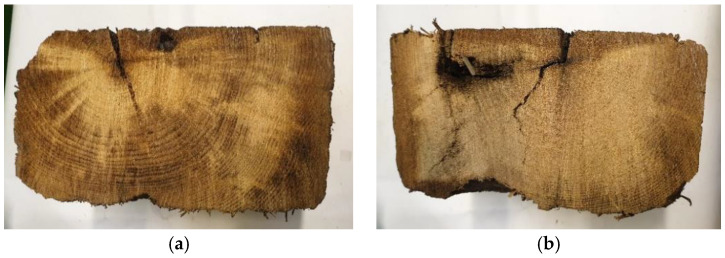
Cross section of the tested tie through the center of length (**a**) and 1 cm from the edge (**b**).

**Figure 2 materials-16-02704-f002:**
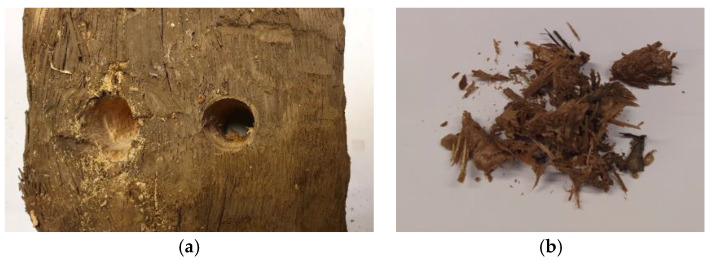
Drilling (**a**) and the obtained raw material sample (**b**).

**Figure 3 materials-16-02704-f003:**
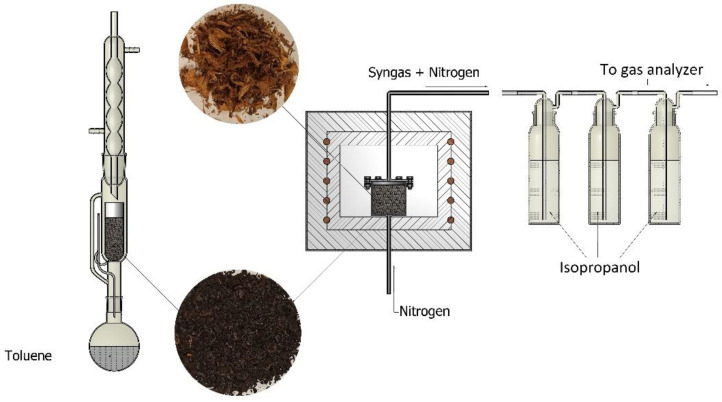
Scheme of the experimental setup for thermal neutralization of waste.

**Figure 4 materials-16-02704-f004:**
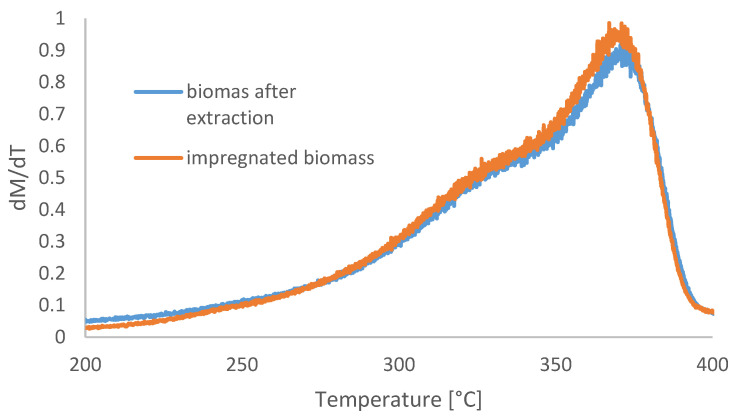
TGA analysis of crude railroad ties and samples of the railroad after extraction.

**Figure 5 materials-16-02704-f005:**
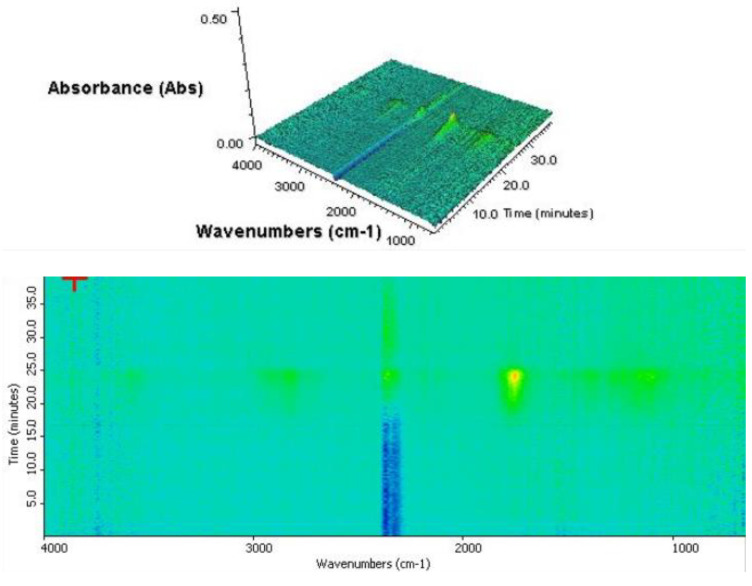
FTIR analysis of the purified wooden tie sample.

**Figure 6 materials-16-02704-f006:**
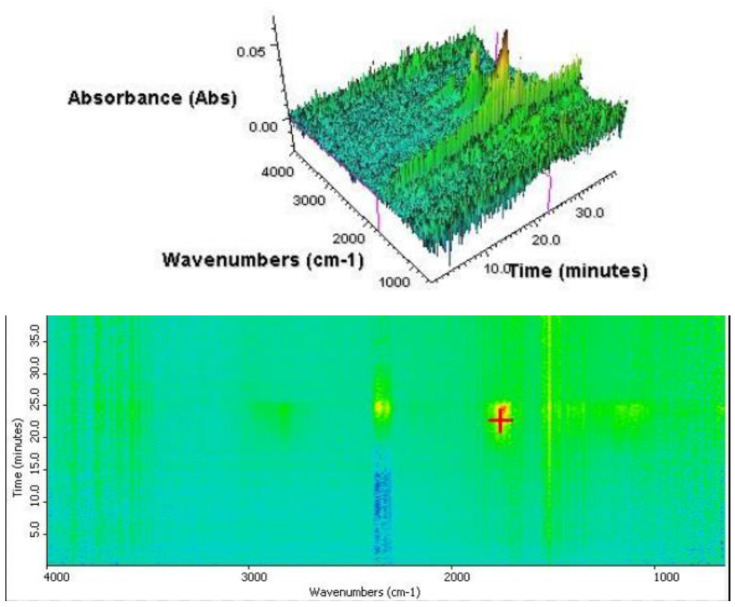
TGA analysis of the impregnated wooden tie sample.

**Figure 7 materials-16-02704-f007:**
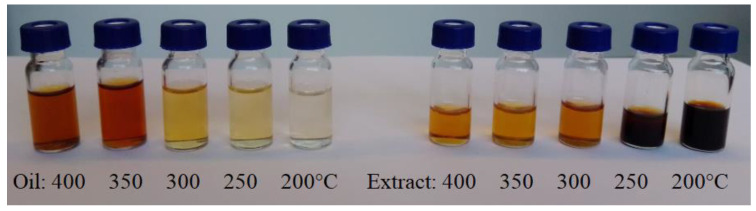
Extracts from the obtained chars (**left**) and pyrolytic oil (**right**).

**Figure 8 materials-16-02704-f008:**
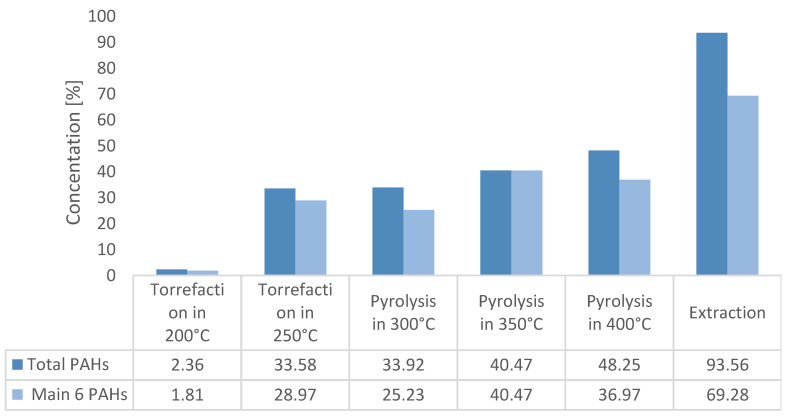
Results of PAH content in the obtained pyrolytic oils.

**Figure 9 materials-16-02704-f009:**
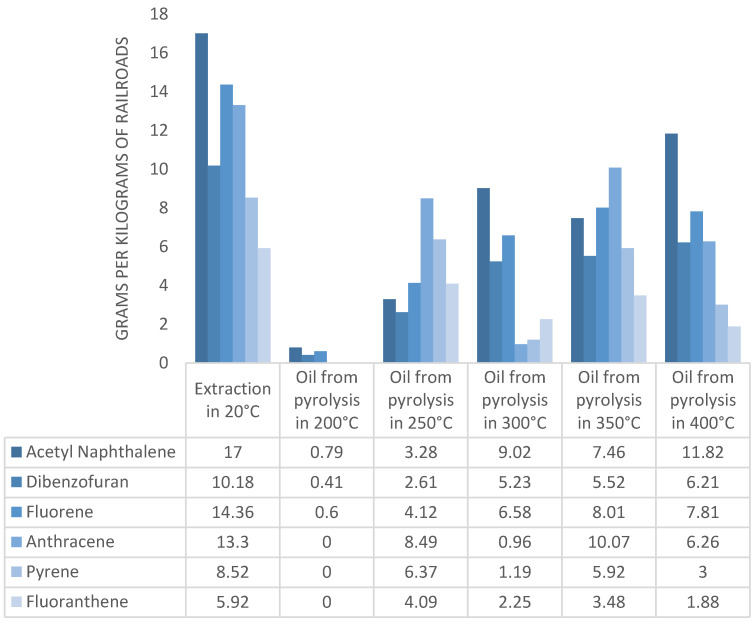
Analyses of pyrolytic oil.

**Figure 10 materials-16-02704-f010:**
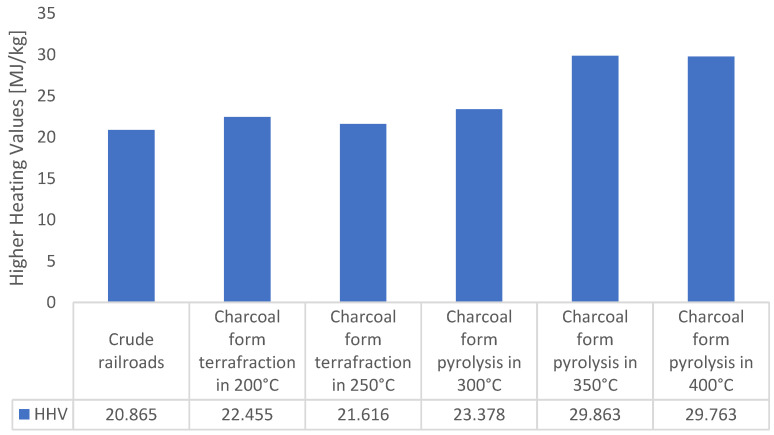
The higher heating value of the raw material (20 °C) and the chars obtained with different final temperatures.

**Figure 11 materials-16-02704-f011:**
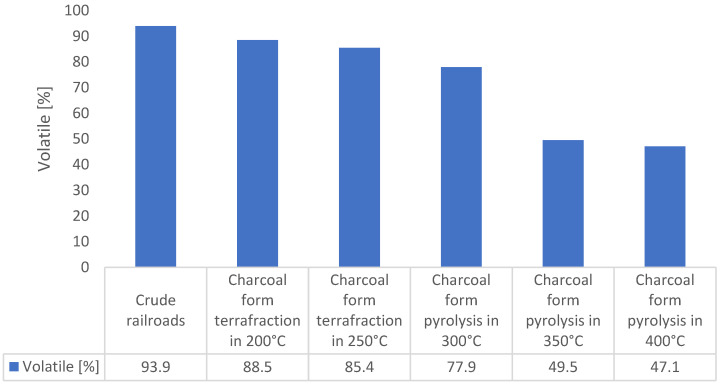
Volatile content of raw material (20 °C) and the chars obtained with different final temperatures.

**Figure 12 materials-16-02704-f012:**
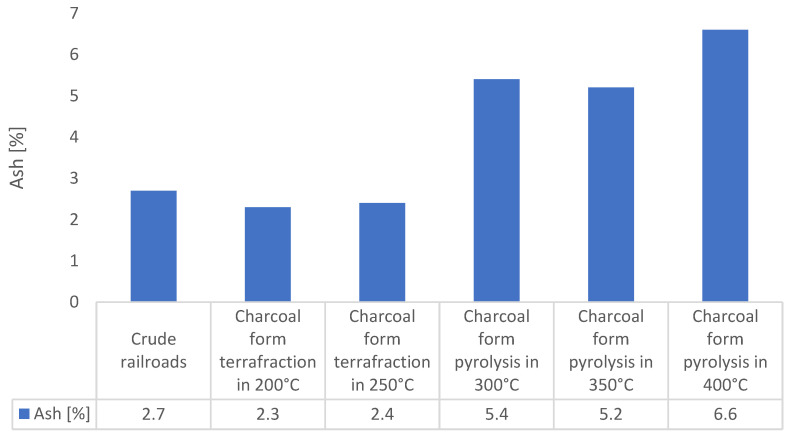
Ash content of raw material (20 °C) and the chars obtained with different final temperatures.

**Table 1 materials-16-02704-t001:** Railway ties in Poland based on [12].

Railway Tie Type	Quantity [mln Pieces of Railroad Ties]	Lifespan [Years]
Softwood	17.8	18
Hardwood	2.3	25
Prestressed Concrete	37	35
Other	1.2	30

## Data Availability

The data presented in this study are available on request from the corresponding author.

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
