# Peer review of "Pyrolysis and Torrefaction—Thermal Treatment of Creosote-Impregnated Railroad Ties as a Method of Utilization"

_materials, 2023, doi:10.3390/ma16072704_

Round 1

Reviewer 1 Report

The manuscript has been upgraded compared to its previous content. It can be accepted for publication. Nevertheless, it still contains multiple editing errors, such as:
Lines 17 and 18 (units);
Line 68 (unit);
Line 45 - did the Authoirs mean 35, 3.5 or 3-5 ?
Line 71 A sentence should start with a capital letter
Line 154 Solvent 159?
etc.
Please double-check to eliminate such issues as they lower the overall soundness of the paper.

Author Response

thank you very much for your review. The answer is in the attachment

Reviewer 2 Report

The scientific article presented certainly required a considerable experimental effort. Unfortunately, it is not clear why the impregnating oil must be extracted if the wooden sleepers are burned afterwards. Wouldn't it be sufficient to have good combustion directly on the sleepers themselves to eliminate the PAHs during combustion?

The results are not presented in a readable manner, and I recommend that the work be presented again, subdividing it into further subsections. There are many scientific explanations missing. For example, at line 136, it is reported that the biomass is dried before treatment. Drying leads to the loss of analytes, and this operation (temperature, duration) has not been described.

At line 147, the operation of the soxhlet is described, which is not necessary in a scientific journal. Instead, the extraction solvent, the extraction time or the minutes required per extraction cycle and the total time should have been described.

At line 166, the thermogravimetry of both the biomass itself and the biomass extracted in the soxhlet is described. This experiment does not make sense, as the extracted biomass is impregnated with the extraction solvent.

Line 262 mentions "Fluoronaphtalene" but it is not clear what this is. Additionally, in Figure 9, "Acetyl Naphthalene" is present. There is a general lack of description of all the analytical methods used, including which analytes were analyzed and which PAHs.

At line 280, it seems anomalous that the removal of aromatic compounds increased the calorific value. Perhaps the treatments removed both the impregnation water and the water bound to the cellulose and lignin?

Author Response

Thank you very much for your review, the answer is attached

Reviewer 3 Report

Dear authors 

I have indicated some minor revisions on the word document in the attachment. Please correct them, then I can recommend to the publication.

Author Response

(The authors gave the same response as above.)

Reviewer 4 Report

The paper need of a linguistic restyling from a mother tongue, in some part that I signed in the attached file the meaning is so clear.

Author Response

(The authors gave the same response as above.)

Reviewer 5 Report

Comments to the authors

The present study describes the Pyrolysis and terrefraction - thermal treatment of creosote impregnated railroad ties as a method of utilization.

In many places, in many paragraphs, no citation was observed.

The manuscript has many typographical errors, interpretation needs to be improved, and the ideas presented in the Ms are not clearly explained and presented in a meaningful manner. The English language/grammar needs to be improved throughout the MS. I regret to inform you that I cannot accept the article in its current form.

Specific comments:

The abstract is not written well, so I suggest you re-write it. The research article is poorly written.

You need the check and clear/explain the highlighted parts in the text in a correct manner to make it understood by the audience. I could not understand the study very well because of the poor interpretation and explanation of the information/data. The references are cited in the text needs to be figured out and corrected.

Results and discussion- This part of the Ms is not focused, scattered and lost. The findings of the study are not supported with the current relevant literature.

I have also given the comments in the annotated document, so check it out and address it correctly and sincerely.

Author Response

(The authors gave the same response as above.)

Round 2

Reviewer 5 Report

I have given the comments in the annotated document. So correct it accordingly.

Author Response

thanks for pointing out where to improve. The latest version of the work is attached
